# In Vivo Study of the Efficacy and Safety of 5-Aminolevulinic Radiodynamic Therapy for Glioblastoma Fractionated Radiotherapy

**DOI:** 10.3390/ijms22189762

**Published:** 2021-09-09

**Authors:** Junko Takahashi, Shinsuke Nagasawa, Motomichi Doi, Masamichi Takahashi, Yoshitaka Narita, Junkoh Yamamoto, Mitsushi J. Ikemoto, Hitoshi Iwahashi

**Affiliations:** 1Graduate School of Information, Production and Systems, Waseda University, Fukuoka 808-0135, Japan; 2Health and Medical Research Institute, National Institute of Advanced Industrial Science and Technology (AIST), Ibaraki 305-8566, Japan; m.ikemoto@aist.go.jp; 3Department of Radiology, Graduate School of Medical Science, Kyoto Prefectural University of Medicine, Kyoto 602-8566, Japan; snaga@koto.kpu-m.ac.jp; 4Biomedical Research Institute, National Institute of Advanced Industrial Science and Technology (AIST), Ibaraki 305-8566, Japan; doi-m@aist.go.jp; 5Department of Neurosurgery and Neuro-Oncology, National Cancer Center Hospital, Tokyo 104-0045, Japan; masataka@ncc.go.jp (M.T.); yonarita@ncc.go.jp (Y.N.); 6Department of Neurosurgery, University of Occupational and Environmental Health, Fukuoka 807-8555, Japan; yama9218@med.uoeh-u.ac.jp; 7The United Graduate School of Agricultural Science, Gifu University, Gifu 501-1193, Japan; h1884@gifu-u.ac.jp

**Keywords:** radiation therapy, fractionated radiotherapy, radiodynamic therapy, glioma, glioblastoma, 5-aminolevulinic acid, protoporphyrin IX, ATPase inhibitory factor 1, U251MG, U87MG

## Abstract

To treat malignant glioma, standard fractionated radiotherapy (RT; 60 Gy/30 fractions over 6 weeks) was performed post-surgery in combination with temozolomide to improve overall survival. Malignant glioblastoma recurrence rate is extremely high, and most recurrent tumors originate from the excision cavity in the high-dose irradiation region. In our previous study, protoporphyrin IX physicochemically enhanced reactive oxygen species generation by ionizing radiation and combined treatment with 5-aminolevulinic acid (5-ALA) and ionizing radiation, while radiodynamic therapy (RDT) improved tumor growth suppression in vivo in a melanoma mouse model. We examined the effect of 5-ALA RDT on the standard fractionated RT protocol using U251MG- or U87MG-bearing mice. 5-ALA was orally administered at 60 or 120 mg/kg, 4 h prior to irradiation. In both models, combined treatment with 5-ALA slowed tumor progression and promoted regression compared to treatment with ionizing radiation alone. The standard fractionated RT protocol of 60 Gy in 30 fractions with oral administration of 120 and 240 mg/kg 5-ALA, the human equivalent dose of photodynamic diagnosis, revealed no significant increase in toxicity to normal skin or brain tissue compared to ionizing radiation alone. Thus, RDT is expected to enhance RT treatment of glioblastoma without severe toxicity under clinically feasible conditions.

## 1. Introduction

Fractionated radiotherapy (RT; 60 Gy in 30 fractions delivered over six weeks) is the standard treatment for malignant glioma, which is generally performed after surgery in combination with temozolomide (TMZ) to improve overall survival. The recurrence rate of malignant glioblastoma is extremely high, and most recurrent tumors originate from the excision cavity in the high-dose irradiation region [1], with a median survival of approximately 14 months. However, a sufficient RT effect was not observed. In RT for invasive cancers such as malignant glioma, it is desirable to contrast the dose of radiation administered to normal cells and cancer cells. RT technologies, such as three-dimensional RT, intensity-modulated RT, proton beam therapy, boron neutron capture therapy (BNCT), and heavy ion beam therapy, have allowed remarkable progress in the spatial distribution of the radiation dose to cancer tissue—sparing normal tissue. However, it is difficult to irradiate only invading cancer cells while excluding any surrounding normal cells. It is disadvantageous to increase the dose by 60 Gy, especially when the brain stem and optic chiasm can only be dosed at approximately 50 Gy, due to the possibility of RT-induced radiation necrosis; the risk of which increased further when combined with TMZ [2,3].

5-Aminolevulinic acid (5-ALA) is a non-fluorescent prodrug that leads to intracellular accumulation of fluorescent porphyrins in malignant gliomas for intraoperative identification and resection of these tumors. The effectiveness of 5-ALA-mediated photodynamic diagnosis (5-ALA-PDD) has been clinically verified and is widely used in current brain tumor resection surgery [4,5]. 5-ALA photodynamic therapy (5-ALA-PDT) has been used to treat precancerous lesions and cancers such as skin cancer [6,7]. For glioma, interstitial PDT (a stereotactic phototherapy using fiber optics as a laser diffuser due to limited light transmission) or intraoperative PDT (which uses a lighting device placed inside the surgical cavity) have been developed [8,9,10]. Clinical studies and clinical trials of 5-ALA PDT for non-dermatologic cancer are ongoing [11,12]. The mechanism of 5-ALA-PDT is as follows: exogenous 5-ALA induces the accumulation of protoporphyrin IX (PpIX) in cancer cells, PpIX produces reactive oxygen species (ROS), mainly ^1^O_2_, under laser light suitable for excitation of PpIX, which induces cellular damage and death [13,14].

Previously, we found that PpIX enhances ROS generation by ionizing radiation and that the major ROS present are the hydroxyl radical (∙OH), superoxide anion (O_2_^−^), and singlet oxygen (^1^O_2_) using ROS detection reagents [15]. Our findings suggest the possibility of RT using PpIX as a radiomediator. We defined RT using a radiomediator, such as PpIX, which reacts physicochemically with ionizing radiation as radiodynamic therapy (RDT) and verified its effectiveness in vitro and in vivo. Our studies using the subcutaneous melanoma model and brain metastasis model showed that 5-ALA pre-treatment enhanced the effectiveness of X-ray irradiation by acting as a radiomediator that facilitates PpIX accumulation in tumors and enhances ROS production [16,17].

Due to advances in glioblastoma prevention, early detection, and treatment, five-year glioblastoma survival has increased from 4% to 7% over the past four decades [18]. The development of treatment methods that aim to prolong life and cure the condition is strongly desired. In a retrospective analysis, RT improved the overall survival of glioblastoma patients with poor performance status [19]. We believe that improving RT is a key issue in controlling glioblastoma. In this study, we examined the cellular response to single-dose irradiation with 5-ALA for two types of glioblastoma and the effect of RDT for the standard fractionated RT treatment protocol for glioblastoma after surgery using the same two types of glioblastoma-bearing mice. We then verified the safety of 5-ALA combined with standard fractionated RT in normal mice. 

## 2. Results

### 2.1. 5-ALA Enhanced Cellular Response to Single-Dose Ionizing Radiation

The combined effects of 5-ALA and ionizing radiation on the human glioblastoma cell lines U251MG and U87MG were examined in vitro using a clonogenic assay (Figure 1). Glioblastoma cells were incubated for 4 h prior to irradiation. Porphyrin accumulation in these cells increased as the 5-ALA concentration increased (Table 1). The results showed that single-dose ionizing radiation suppressed the growth of glioblastoma cells in a dose-dependent manner, and 5-ALA enhanced these effects. It is thought that 5-ALA-induced PpIX enhanced ionizing radiation damage in both glioblastoma cell lines. U251MG cells tended to have higher intracellular porphyrin concentrations than U87MG cells, but there was little difference between the two cell lines depending on the X-ray dose and 5-ALA concentrations for the survival fraction.

### 2.2. 5-ALA Enhanced the Response to Fractionated Irradiation in Terms of Tumor Growth

To evaluate the response of tumors to 5-ALA treatment with fractional irradiation in vivo, two glioblastoma tumor models, U251MG and U87MG, were used. After the tumor volume reached 150–200 mm^3^ in U251MG-bearing mice or 180–220 mm^3^ in U87MG-bearing mice, groupings were performed and two series of irradiations were initiated. The tumor was irradiated with fractionated irradiation at a total dose of 60 Gy (2 Gy daily × 5 days/week × 6 weeks). 5-ALA was orally administered at 60 or 120 mg/kg, 4 h before each irradiation. The dose and timing of ionizing irradiation were based on the clinical conditions of RT in clinical patients. The timing of 5-ALA administration was the same as that in the clinics. The doses of 5-ALA were less than those used in clinical conditions of PDD, considering the human equivalent dose (240 mg/kg for mice). 5-ALA is administered as a single dose in PDD, whereas a repeated dose is administered during X-ray irradiation. Therefore, we set a lower concentration per dose than the concentration of a single dose of PDD. Five groups were created for this experiment: NT (total X-ray treatment, 0 Gy; 5-ALA treatment, 0 mg/kg/day), 120ALAT (0 Gy, 120 mg/kg/day), XT (2 Gy/day, total 60 Gy, 0 mg/kg/day), 60ALAXT (60 Gy, 60 mg/kg/day), and 120ALAXT (60 Gy, 120 mg/kg/day) groups.

In the NT and 120ALAT groups, tumors grew in U251MG-bearing mice treated without X-ray irradiation (Figure 2A). Mice were euthanized when the tumor size reached ≥400 mm^3^. In the U251MG XT group, the tumors grew slightly up to four weeks after the initiation of X-ray irradiation, after which they regressed when irradiation was continued. Tumor regression was observed after two weeks in the 60ALAXT group. In contrast, tumor regression was observed from the first week in the 120ALAXT group of U251MG-bearing mice (Figure 2A). In the NT and 120ALAT groups, tumors grew in U87MG-transplanted mice treated without X-ray irradiation (Figure 2B). The tumor size decreased from the first week after initiation of X-ray irradiation in the XT, 60ALXT, and 120ALAXT groups and all X-ray irradiation groups of U87MG-bearing mice (Figure 2B). From four to six weeks, the tumor size in all X-ray irradiation groups was very small, and there were no differences between the groups. Therefore, we continued to observe the tumor size. After seven weeks, tumor regrowth was observed in the XT group, but not in the 60ALAXT and 120ALAXT groups of U87MG-bearing mice (Figure 2B). Porphyrin accumulation in U251MG and U87MG tumors treated with 5-ALA for 4 h increased as the 5-ALA concentration increased, and there was little difference between the two cell lines (Table 2). No significant body weight changes were observed in the U251 cancer-bearing mice during the irradiation period (Appendix A). However, U87 cancer-bearing mice showed a slight decrease in body weight at 10 weeks (Appendix A).

### 2.3. Morphological Observation of Tumor Tissues

The morphological characteristics of the tumors were obtained by hematoxylin–eosin (HE) staining, mitochondria, and nuclei staining. ATPase inhibitory factor 1 (ATPIF1) immunostaining was performed for mitochondrial observation, and Hoechst^®^ 33342 staining was performed for nuclear observation (Figure 3). The NT and 120ALAXT groups had uniform nuclei, while cells with enlarged nuclei were observed in the XT group in both cell lines. The 120ALAXT group of both cell lines showed shrunken nuclei with a decrease in the number of nuclei, suggesting enlargement of cells due to abnormal cell division. ATPIF1 was observed to be stained in the nucleus in both cell lines, suggesting a concentration around the nucleus in the 120ALAXT group.

### 2.4. Effect of 5-ALA on Fractionated Irradiation in Normal Tissues

The effects of 5-ALA administration on ionizing radiation were examined in normal animals, concentrating on the brain. Five groups were created for this experiment: NT (total X-ray treatment, 0 Gy; 5-ALA treatment, 0 mg/kg/day), 240ALAT (0 Gy, 240 mg/kg/day), XT (2 Gy/day, total 60 Gy, 0 mg/kg/day), 120ALAXT (60 Gy, 120 mg/kg/day), and 240ALAXT (60 Gy, 240 mg/kg/day) groups. During the irradiation period, the body weight (Appendix A) and the conditions of the animals were observed. In the NT and 240ALAT groups, a tendency toward an increase in body weight was observed. In groups XT, 120ALAXT, and 240ALAXT, a slight decrease in body weight was observed from two weeks after the initiation of X-ray irradiation; however, there was no significant difference found due to 5-ALA administration. All mice were also examined for their activity during the treatment period. No neurological signs (seizure, crouching posture, or abnormal behavior) were observed.

After the completion of 60 Gy fractional irradiation combined with 5-ALA treatment, pathological evaluation of the irradiation site of the skin and brain was performed. Blood was collected and tissues of the skin and brain were fixed 24 h after the final 5-ALA administration. The safety test protocol is illustrated in Figure 4.

Pathological evaluation of the skin of all X-ray irradiated mice revealed the following: (1) hypertrophy (mild to moderate) of epidermal or epidermal cells, (2) hypertrophy (mild) of hair follicle epithelial cells, (3) apoptosis (mild) of epidermal cells, (4) atrophy/disappearance (mild) of hair follicles, and (5) atrophy/disappearance (moderate) of epidermal cells. However, these skin lesions were not different among the X-ray irradiation groups due to 5-ALA administration (Figure 5 and Table 3). Pathological evaluation of the brain showed no changes associated with X-ray irradiation or 5-ALA administration (Table 3). In the normal brain, 5-ALA does not pass through the blood brain barrier [20,21], which is consistent with our pathological examination results.

Serum biochemical tests were also performed (Appendix A) to evaluate the levels of total protein, albumin, albumin/globulin ratio, urea nitrogen, creatinine, Na, K, Ca, IP, AST, ALT, ALP, total cholesterol, neutral fat, total bilirubin, and blood glucose. Uric acid levels tended to be lower in all X-ray irradiation groups. AST ALT levels tended to be higher in the 5-ALA treatment group. IP and ALP levels were significantly lower in the X-ray treatment group than in the NT group. However, this value increased with the combined use of 5-ALA (Appendix A). 

## 3. Discussion

The standard of care for patients with newly diagnosed glioblastoma is maximal surgical resection followed by RT with concurrent and adjuvant TMZ chemotherapy (TMZ-CHT). The standard protocol is fractionated focal irradiation in daily fractions of two Gy administered five days per week for six weeks, up to a total of 60 Gy. To date, high-dose RT at a total dose of 90 Gy (90 Gy in 45 Fr) has led to a remarkable improvement in outcomes [22,23]. Hypofractionated RT (40 Gy in 15 Fr or 60 Gy in 15 Fr) has led to a reduction in the burden of RT on the elderly [24,25]. In most cases, local recurrence originated from the excision cavity in the high-dose irradiation region [1,22,23]. Therefore, we verified the effectiveness and safety of 5-ALA RDT using a standard glioblastoma RT protocol. 

Previously, we attempted to detect the physicochemical reaction between PpIX and X-rays using ROS indicators [15]. To identify the ROS, 2-[6-(4-amino)phenoxy-3H-xanthen-3-on-9-yl] benzoic acid (APF), which mainly detects ∙OH, and dihydroethidium (DHE), which detect O_2_^−^ and/or ^1^O_2_, were used together with ethanol as an ∙OH quencher. The results indicated that PpIX contributes to enhanced generation of ∙OH, O_2_^−^, and/or ^1^O_2_ under X-ray irradiation, but to O_2_^−^ and/or ^1^O_2_ under UV irradiation. PpIX was excited by UV light. Although the types of ROS produced are different, it has been shown that a reaction similar to photoexcitation is triggered by X-rays [15]. We termed our treatment regimen RDT. Radiodynamic therapy (RDT) may be confused with photodynamic therapy (PDT) in appearance, but I use the term “RDT” intentionally. We are not the first to use this term; “RDT” was previously used to mean RT using the drug for PDT, but no direct data on the physicochemical reaction between ionizing radiation and sensitizers has been reported [26,27]. Therefore, we defined RDT as using a radiomediator, such as PpIX, which physicochemically enhances ROS generation by ionizing radiation based on experimental verification [15]. Several groups have studied the effectiveness of 5-ALA RDT in vitro and in vivo. In 2009, the physicochemical reaction of PpIX and X-rays, which enhanced the production of ROS, was reported [15]. We have previously reported the effectiveness of RDT using not only an X-ray irradiator with 160 kV nominal X-ray tube voltage, but also a medical linear accelerator in a mouse melanoma tumor model [16,17,28,29]. Yamamoto et al. demonstrated the effectiveness of RDT for gliomas in vitro and in vivo [30,31,32,33]. They also performed immunohistochemical analysis to reveal that numerous ionized calcium-binding adapter molecule 1-positive macrophages gathered at the surface and within the subcutaneous tumors [32]. RDT using the glioma stem cell brain transplant model [34], colorectal cancer transplant model [35], and prostate cancer transplant model have been reported [36]. The effectiveness of RDT using a high-energy (15-MV) linear accelerator [37] and carbon-ion beam irradiation [38] has also been reported.

Cancer cells were transplanted under the skin to evaluate the tumor regression. This is because the subcutaneous model makes it easier to observe tumor size, and large intracranial tumors affect mouse survival. There are established systems that measure the growth of cancer by fluorescence intensity using a xenograft with a cell line stably integrated with a luciferase reporter gene. We used cell lines with luciferase when the tumor diameter became undetectable during the experiment. U251MG and U87MG cell lines are widely used in glioblastoma studies [39,40,41]. In this study, tumor growth was observed in U251MG -bearing mice treated with X-ray irradiation alone for three weeks. However, suppression of tumor growth was observed in U87MG-bearing mice at the beginning of X-ray irradiation. In U251MG- and U87MG-bearing mice, combined treatment with 5-ALA and X-ray was more effective in suppressing tumor growth or tumor regression than X-ray alone. There was little difference in the intracellular accumulation of 5-ALA-induced PpIX or the survival fraction between U87MG and U251MG cells in vitro. The difference in tumor growth between the cell lines in response to X-ray irradiation may be attributed to the difference in the cell growth rate. There were no significant differences found between the characteristics of U251MG and U87MG in other studies [39,40,41]. The tumor regressed faster in the U87MG group than in the U251MG with X-ray irradiation alone and combined with 5-ALA, and irradiation was continued even after the tumor regressed in this study. After surgical and adjuvant therapy, reoccurrence at the periphery of the tumor cavity resulted in poor survival of glioblastoma. Therefore, suppressing regrowth may lead to an improved survival rate. Kim et al. examined the genes involved in tumor regrowth by microarray analysis of tumor tissue and identified highly expressed genes associated with carcinogenesis, such as TMEM119, FST, RSPO3, ROS1, and NBL1 [42]. Xie et al. reported that activation of ERK1/2 by STAT3 inhibition is responsible for the survival of radioresistant tumor cells, and dual inhibition of ERK1/2 and STAT3 inhibits tumor regrowth [43]. Singer et al. showed that inhibition of self-renewal of glioma stem cells was mediated by downregulation of the stem cell regulators Sox2, Id1, and p-STAT3 [44]. Molecular biological analysis of cell regrowth after irradiation and therapeutic strategies targeting the suppression of regrowth are needed in the future to further verify results.

Systemic administration of 5-ALA leads to the accumulation of PpIX in tumor cell mitochondria as the main photoactive product PpIX [7]. Mitochondria are thought to be damaged by ROS caused by the physicochemical reaction of porphyrins and radiation in RDT. However, the effect of RDT on mitochondrial membrane potential was not observed [17]. Therefore, ATPIF1 immunostaining was performed to observe mitochondrial morphology. ATPIF1 has been confirmed to show almost the same distribution as MitoTracker [45]. Changes in the intracellular distribution of ATPIF1 were observed in the 5-ALA and X-ray combination groups. ATPIF1 is considered a driving oncogene, and overexpression of ATPIF1 has been observed to promote tumor growth and metastasis in tumor cells [46]. Wu et al. showed that ATPIF1 knockdown reduced the migratory and invasive abilities of U251 and U87 cells. They also showed that the presence of ATPIF1 expression was associated with a reduced overall survival rate in patients with glioma [46]. Thus, although the expression of ATPIF1 is related to the growth of cancer, its distribution has not been examined, and a more detailed examination is needed. 

Since a research group at the National Medical Laser Center in London treated patients with laryngeal cancer by oral administration of 5-ALA to accumulate PpIX in oral cavity squamous cell carcinomas in a site-specific manner in 1993 [47], oral administration has been performed for treatment. Furthermore, 5-ALA did not induce phototoxicity compared to other photosensitizers. Oral administration routes and low phototoxicity are critical advantages for the clinical use of drugs, especially with standard fractionation RT. When 5-ALA is used in combination with RT, it is necessary to administer 5-ALA before daily irradiation, and oral administration allows patients to take the drug in advance by themselves. Although 5-ALA is advantageous for clinical use, the safety of repeated administration is a concern. Since PDD or PDT is a single treatment, 5-ALA is administered in a single dose. However, the standard treatment for brain tumors can be as long as 6 weeks, and the safety of repeated administration during this period is not guaranteed. In particular, the brain is an important organ, and the effects of combined treatment with 5-ALA and X-rays are unknown. Therefore, we verified the safety of treatment of the brain and skin in normal animals. The tested concentrations of 5-ALA were 120 mg/kg—a low dose sufficient to show the effectiveness of RDT in this study—and 240 mg/kg. We utilized a guidance document from the US Department of Health and Human Services to determine dosages, which was the Food and Drug Administration’s (FDA) “Industry Guidance for Estimating the Maximum Safe Starting Dose in Initial Clinical Trials for Healthy Adult Volunteer Therapeutics” [48]. FDA guidance recommends to evaluate the available animal data so that a “Non Observed Adverse Effect Level (NOAEL)” can be determined and calculate human equivalent dose (HED). For mice, HED multiplies the animal dose by 0.08 [48]. Therefore, 240 mg/kg is 19.2 mg/kg for humans, which is roughly equivalent to the human dose of 20 mg/kg for PDD. Pathological examination of the skin of all 60 Gy fractional RT-treated mice showed hypertrophy of the epidermal and hair follicle epithelial cells, apoptosis of the epidermal cells, and atrophy/disappearance of hair follicles and epidermal cells. However, there were no differences observed among the XT, 120ALAXT, and 240ALAXT groups, and the degree was mild or moderate. No changes were observed to be associated with X-ray irradiation or 5-ALA administration in the brain. Biochemical studies of the serum showed that repeated doses of high concentrations of 5-ALA could cause liver damage. The administration of 5-ALA or the combined effect of 5-ALA, anesthesia, and tumor resection can cause a mild and reversible elevation in liver enzymes [49]. However, further continuous safety evaluations are required for practical clinical use.

## 4. Materials and Methods

### 4.1. Cell Culture

U251MG glioblastoma cells (U-251 MG-Luc) were purchased from the Japanese Collection of Research Bioresources (Osaka, Japan). U87MG cells (U-87 MG-Luc2) were kindly provided by Dr. Takahashi, National Cancer Center Hospital (Tokyo, Japan). Both cell lines were cultured in Eagle’s minimum essential medium (FUJIFILM Wako PureChemical Corporation; Osaka, Japan) supplemented with 10% FBS (FUJIFILM Wako) in a 5% CO_2_ humidified incubator at 37 °C. The medium was supplemented with 100 units/mL penicillin and 100 μg/mL streptomycin (FUJIFILM Wako). 

### 4.2. X-ray Irradiation Conditions

X-ray irradiation was performed using an MBR-1520R-4 irradiator (Hitachi Power Solutions; Hitachi, Japan) with X-ray energy outputs of 150 kV and 20 mA, with added filtration of 0.5 mm Al and 0.1 mm Cu. The dose rate was 1.0 Gy/min at the sample stage.

### 4.3. Clonogenic Assay

Cells were incubated in complete medium containing 0, 30, or 100 μg/mL 5-ALA (FUJIFILM Wako) for 4 h and then washed with PBS. The medium was replaced with fresh culture medium, and the cells were irradiated at a dose rate of 1 Gy/min. After exposure to 0–4 Gy of X-ray radiation, the cells were seeded in 25-cm^2^ culture flasks with a seeding density of 1000 cells/flask and incubated at 37 °C with 5% CO_2_. The surviving fractions were determined using clonogenic assays. Colonies were fixed and stained with crystal violet (2% in methanol) for at least 14 days after subculturing. Only colonies formed from more than 50 cells were considered.

### 4.4. RDT Treatment for Two Types of Glioblastoma-Bearing Mice 

Subcutaneous implantation of U251MG and U87MG glioma

Human glioblastoma U251MG or U87MG cells stably integrated with a luciferase reporter gene were used as subcutaneous xenografts. Six-week-old female nude mice (BALB/c nu/nu, Charles River Laboratories Japan, Inc.; Yokohama, Japan) were anesthetized and 1.0 × 10^6^ U251MG or U87MG cells were subcutaneously injected into the head. 

#### 4.4.1. Grouping and RDT Fractionated Irradiation

After the tumor volume reached 150–200 mm^3^ in U251MG-implanted mice and 180–220 mm^3^ U87MG-implanted mice, mice were divided into five groups to ensure tumor volume uniformity: (1) NT, control group (*n* = 4); (2) 120ALAT, 120 mg/kg ALA was administered orally (*n* = 4); (3) XT, total dose of 60 Gy (2 Gy daily q.d. × 5 days/week × 6 weeks) (*n* =5); (4) 60ALAXT, 60 mg/kg 5-ALA was administered orally 4 h before each irradiation (*n* = 5); and (5) 120ALAXT, 120 mg/kg 5-ALA was administered orally 4 h before each irradiation (*n* = 5). For X-ray irradiation, a mouse was tightly held in a plastic holder with an opening above the tumor area. The collimated X-ray beam irradiated a 20 × 20 mm area of the brain at the tumor site, which was sufficiently large to cover the entire tumor. Mice in the ALA and X-ray treatment groups were orally administered 5-ALA diluted in PBS at 0, 60, and 120 mg/kg body weight 4 h before X-ray irradiation. Mice treated without X-ray irradiation received 0 or 120 mg/kg 5-ALA at the same time. For U251MG-transplanted mice, the mice were sacrificed one day after the last 30 sessions of X-ray irradiation. U87MG-transplanted mice were sacrificed 4 weeks after the last 30 sessions of X-ray irradiation. For the no X-ray irradiation groups, mice were sacrificed after the tumor volume reached 500 mm^2^. The tumors were weighed using an electronic balance. 

#### 4.4.2. Evaluation during the Irradiation Period

Tumor volume, based on caliper measurements, was calculated every week according to the following formula: tumor volume = the shortest diameter^2^ × the largest diameter × 0.5 [50]. Tumor growth was verified using in vivo bioluminescence imaging with the IVIS 100 system (Caliper Life Sciences; Hopkinton, MA, USA), as previously described [8]. Body weight and tumor size were recorded weekly after the transplantation. 

### 4.5. Determination of PpIX Concentration in Cells and Tissue

PpIX in cultured cells or tissue samples was isolated in 50 μL of NaOH (0.1 M) and homogenized on ice using Powermasher II (Assist; Tokyo, Japan). An aliquot (10 μL) of the NaOH-treated sample was withdrawn and used for a protein concentration assay (modified Lowry protein assay kit, Thermo Scientific; Rochford, IL, USA), while the remaining 40 μL of the NaOH-treated sample was denatured by the addition of 150 μL of N,N-dimethylformamide:isopropanol (100:1, *v*/*v*) solution. After overnight incubation, the prepared samples were centrifuged at 12,000 rpm for 10 min. Porphyrin concentration was determined by spectrophotometry at the Soret maximum (405 nm), and fluorescence was measured using an excitation wavelength of 405 nm and an emission wavelength of 635 nm [51].

### 4.6. Immunocytochemistry or Morphological Observation of Tumor Tissues

Tissues were fixed in 4% paraformaldehyde and embedded in paraffin. Paraffin-embedded tissue sections were cut into standard 6-μm sections, deparaffinized in xylene, and rehydrated using graded alcohol solutions. Antigen retrieval was performed for 10 min at 92 °C in EDTA (10 mmol/L, pH 8.0) in a water bath. Endogenous peroxidases were inactivated by immersing the sections in 0.3% hydrogen peroxide for 12 min. The sections were blocked with 5% goat serum for 60 min at 37 °C. The slides were incubated overnight with the primary antibodies at 4 °C. Subsequently, the slides were treated with an appropriate Cy3-conjugated secondary antibody for 60 min at 37 °C. DNA was counterstained in all immunofluorescence experiments using Hoechst^®^ 33258 and is depicted in blue. Finally, the tissues were imaged using a laser confocal microscope (Nikon AI confocal scanning laser microscope, Nikon Instruments Inc.; Tokyo, Japan) with a 60x oil-immersion objective lens.

### 4.7. Safety Test of 5-ALA on Fractionated Irradiation

#### 4.7.1. Mice and Breeding Condition

In this study, 6-week-old female BLBL/c mice (*n* = 23; body weight, 16–21 g; Charles River Laboratories Japan, Inc., Yokohama, Japan) were used. The mice were housed in metal cages with free access to food and water under natural light at 25 °C ± 1 °C and 50%  ±  10% relative humidity at the experimental animal center. The mice were bred under normal light conditions and no special treatments, such as shading, were performed. 

#### 4.7.2. Grouping and RDT Fractionated Irradiation

Five groups were created for this experiment: NT (total X-ray treatment, 0 Gy; 5-ALA treatment/day, 0 mg/kg/day), 240ALAT (0 Gy, 240 mg/kg/day), XT (2 Gy/day, total 60 Gy, 0 mg/kg/day), 120ALAXT (60 Gy, 120 mg/kg/day), and 240ALAXT (60 Gy, 240 mg/kg/day) groups. Mice in the X-ray and ALA groups and all X-ray irradiation groups were irradiated with 2 Gy daily q.d.  × 5 × 6 weeks, for a total dose of 60 Gy, using the same procedure as that used for the mouse tumor models. For X-ray irradiation, a mouse was tightly held in a plastic holder with an opening above the head, and the collimated X-ray beam irradiated a 20 × 20 mm area of the brain. Mice in the XT, 120ALAXT, and 240ALAXT groups were orally administered 5-ALA diluted in PBS at 0, 120, or 240 mg/kg body weight 4 h before X-ray irradiation. Mice in the NT and 240ALAT treatment groups received 5-ALA diluted in PBS at the same time, but did not receive irradiation. Body weights were measured weekly during the experimental period.

#### 4.7.3. Evaluation after RDT Fractionated Irradiation

Mice were sacrificed 1 day after the last 30 sessions of X-ray irradiation, specifically 24 h after the last 5-ALA administration. Serum samples from each mouse were collected for the biochemical tests. Biochemical tests were conducted by Oriental Yeast Co., Ltd. (Nagahama, Japan) using standard clinical methods. The head skin and brain were fixed in 4% paraformaldehyde and subsequently embedded in paraffin. The morphological characteristics of the normal tissues were obtained by HE staining. Pathologists examined these tissues by light microscopy. 

### 4.8. Statistics

Clonogenic assay, tumor growth, body weight, and serum biochemical tests were analyzed using one-way factorial ANOVA, followed by the Tukey–Kramer multiple comparisons test. The Games–Howell post hoc test was used in cases where the variance was not homogenous. Differences were considered statistically significant at *p* < 0.05. 

## 5. Conclusions

This study examined the 5-ALA RDT effect for the standard fractionated RT protocol (60 Gy in 30 fractions delivered over 6 weeks) using U251MG- or U87MG-bearing mice. 5-ALA was orally administered at 60 mg/kg or 120 mg/kg, one-quarter or one-half of the human equivalent dose of PDD, 4 h prior to each irradiation. In both models, combined treatment with 5-ALA slowed tumor progression and promoted regression compared to X-ray irradiation alone. Examination of the effects of the standard fractionated RT protocol with oral administration of 120 and 240 mg/kg 5-ALA revealed no significant toxicity to the normal skin or the brain compared to X-ray irradiation alone. Thus, RDT is expected to enhance the RT of glioblastoma without severe toxicity.

## Figures and Tables

**Figure 1 ijms-22-09762-f001:**
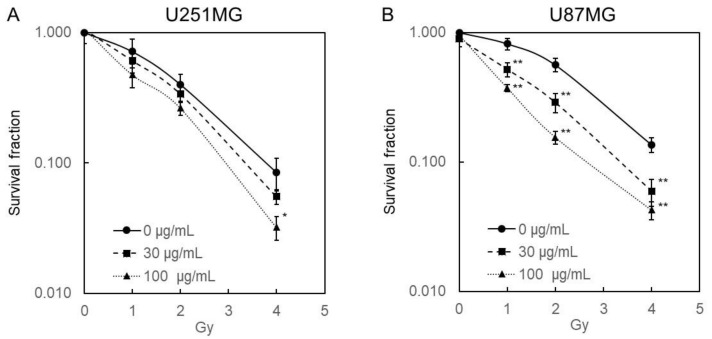
Radiation dose-response curves of (**A**) U251MG and (**B**) U87MG. Both cell lines were exposed to medium containing 0, 30, and 100 μg/mL 5-ALA for 4 h. After the medium was replaced with fresh medium, cells were irradiated at a dose rate of 1 Gy/min. Data are presented as means ± standard deviations (*n* = 3). Statistical significance relative to the experiment performed at the same radiation dose is indicated by (* *p* < 0.05, ** *p* < 0.01).

**Figure 2 ijms-22-09762-f002:**
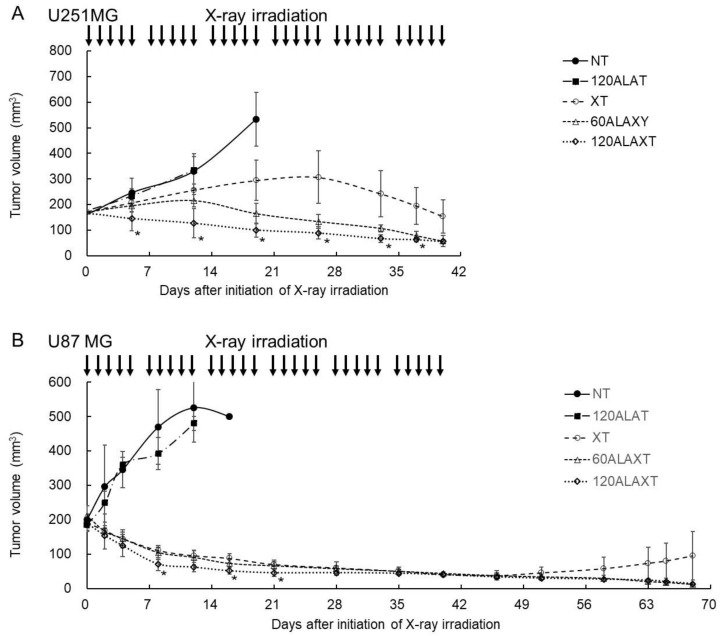
5-ALA potentials for (**A**) U251MG and (**B**) U87MG tumor suppression by fractionated irradiation. After tumor volume reached 150–200 mm^3^ in U251MG-bearing mice or 180–220 mm^3^ in U87MG-bearing mice, X-ray irradiation was initiated. Each U251MG- or U87MG-implanted BALB/c nu/nu mice were divided into five groups: NT, no treatment; 120ALAT; 120 mg/kg/day ALA was administrated orally; XT, total dose of 60 Gy (2 Gy daily × 5 days/week × 6 weeks); 60ALAXT; 60 mg/kg/day 5-ALA was administered orally 4 h before each irradiation; 120ALAXT, 120 mg/kg/day 5-ALA was administered orally 4 h before each irradiation. Data are represented as means ± standard deviations (*n* = 4 or 5). Statistical significance relative to the experiment performed at the same radiation dose is indicated by (* *p* < 0.05).

**Figure 3 ijms-22-09762-f003:**
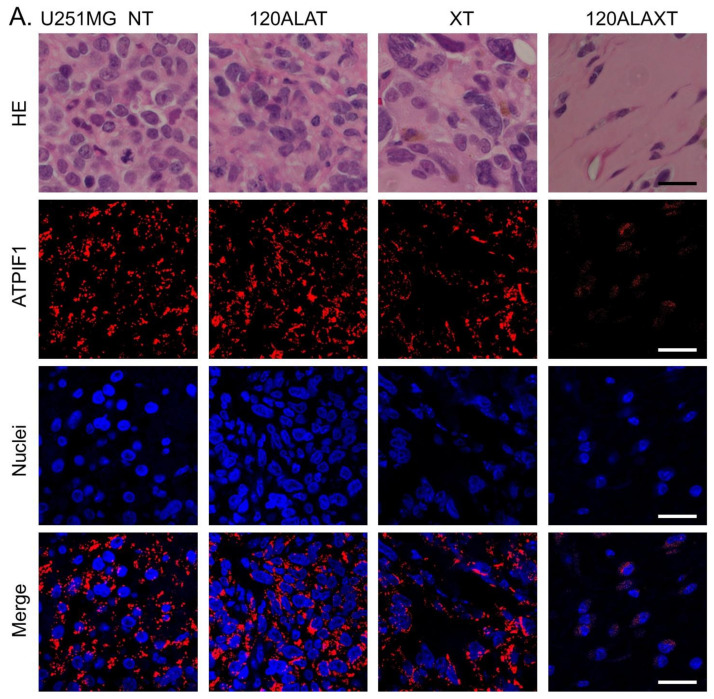
Morphology of tumor sections observed after HE, ATPIF1, and nuclei staining in (**A**) U251MG- and (**B**) U87MG-bearing mice. U251MG-implanted mice were sacrificed one day after the last X-ray irradiation. U87MG-implanted mice were sacrificed 4 weeks after the last X-ray irradiation. ATPIF1 immunostaining was performed for mitochondrial observation and Hoechst^®^ 33342 staining was performed for nuclear observation. The 120ALAXT group in both cell lines shows a decrease in the number of nuclei. ATPIF1 in both cell lines is stained at the nucleus in 120ALAXT group, suggesting that ATPIF1 is concentrated around the nucleus. Bars represent 20 μm.

**Figure 4 ijms-22-09762-f004:**
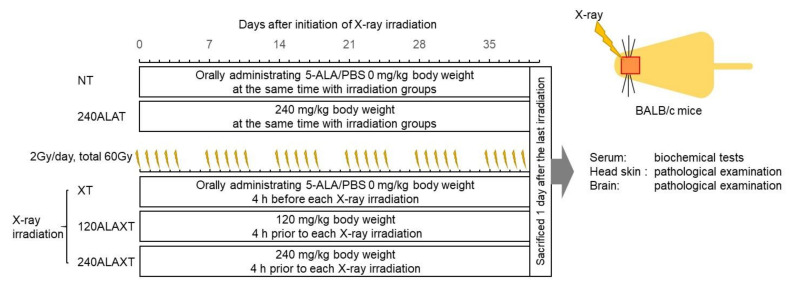
The safety test procedure.

**Figure 5 ijms-22-09762-f005:**
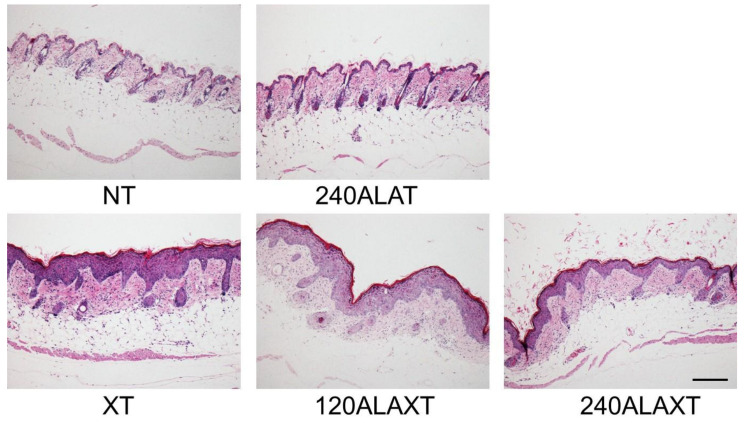
Pathological examination of the effects of ionizing radiation and 5-ALA treatment on normal skin. BALB/c mice were sacrificed one day after the last X-ray irradiation. The skin sections were stained with HE. Bars represent 200 μm.

**Table 1 ijms-22-09762-t001:** Intracellular concentrations of PpIX incubated with 5-ALA for 4 h (*n* = 4).

5-ALA (mg/mL)	U251MG (pM/mg Protein)	U87MG (pM/mg Protein)
0	0.3 ± 0.1	0.2 ± 0.1
30	7.1 ± 1.3	6.7 ± 1.2
100	23.6 ± 5.1	17.8 ± 1.7

**Table 2 ijms-22-09762-t002:** Intratumor concentration of PpIX (*n* = 4).

5-ALA (mg/kg)	U251MG (pM/mg Protein)	U87MG (pM/mg Protein)
0	0.2 ± 0.1	0.3 ± 0.1
60	1.1 ± 0.7	1.6 ± 0.7
120	4.9 ± 0.6	3.5 ± 0.7

**Table 3 ijms-22-09762-t003:** Pathological examination of the safety test.

Group		NT	240ALAT	XT	120ALA XT	240ALA XT
X-ray irradiation		0 Gy	0 Gy	60 Gy	60 Gy	60 Gy
5-ALA (mg/kg/day)		0	240	0	120	240
*n*		4	4	5	5	5
Skin	Pathological findings	N	N	*p*	*p*	*p*
	Acanthosis/fleshiness of epidermal cells	-	-	+++	+++	+++
	Apoptosis of epidermal cells	-	-	+	+	+
	Atrophy or elimination of follicles of pile	-	-	++	++	++
	Atrophy or elimination of sebaceous gland	-	-	+++	+++	+++
Brain	Pathological findings	N	N	N	N	N

Pathological grade: -, unchanged; +, insignificant; ++, light; +++, moderate; *p*, observations positive; N, negative.

## Data Availability

Data is contained within the article or Appendix A.

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
