# Peer review of "In Vivo Study of the Efficacy and Safety of 5-Aminolevulinic Radiodynamic Therapy for Glioblastoma Fractionated Radiotherapy"

_ijms, 2021, doi:10.3390/ijms22189762_

Round 1

Reviewer 1 Report

The authors describe the slowed tumor progression in glioblastoma models with treatment with 5–ALA radiodynamic therapy. 

Given the poor prognosis in glioblastoma, especially in the recurrent setting, improvement of standard therapy is of utmost importance. 

I felt that the studies demonstrating the effects 5-ALA and radiotherapy were sound. 

I only have minor remarks.

Title: The title is somehow misleading, since the study was done in mice, not in humans. The authors should state clearly in the title that the study was performed in mice. 

Introduction: The authors describe the mechanism of 5-ALA PDT, but do not cite the source that clarified the mechanism. The source should be cited. 

Discussion: The sentences about using the term “RDT“ is confusing. The authors should clearly describe their definition of “RDT“. 

Author Response

First of all, we would like to express our cordial thanks for the constructive comments received from this reviewer.

 >The authors describe the slowed tumor progression in glioblastoma models with treatment with 5–ALA radiodynamic therapy.

Given the poor prognosis in glioblastoma, especially in the recurrent setting, improvement of standard therapy is of utmost importance.

I felt that the studies demonstrating the effects 5-ALA and radiotherapy were sound.  

Thank you for your high evaluation for our paper.

I only have minor remarks.

Title: The title is somehow misleading, since the study was done in mice, not in humans. The authors should state clearly in the title that the study was performed in mice. 

Thank you for the comment. In response to the reviewer’s comment, we have changed the title as follows.

In vivo evaluation of the efficacy and safety of 5-aminolevulinic radiodynamic therapy for glioblastoma fractionated radiotherapy

Introduction: The authors describe the mechanism of 5-ALA PDT, but do not cite the source that clarified the mechanism. The source should be cited.

In response to the comments, we added the following references.

(Introduction, Page 2, Line 71)

  1. Castano, A. P.; Demidova, T. N.; Hamblin, M. R. Mechanisms in photodynamic therapy: part one-photosensitizers, photochemistry and cellular localization. Photodiagnosis Photodyn Ther. 2004, 1(4), 279–293.
  2. Berg, K.; Selbo, P. K.; Weyergang, A.; Dietze, A.; Prasmickaite, L.; Bonsted, A.; Engesaeter, B. Ø.; Angell-Petersen, E.; Warloe, T.; Frandsen, N.; Høgset, A. Porphyrin-related photosensitizers for cancer imaging and therapeutic applications. J Microsc. 2005, 218(Pt 2), 133–147.

Discussion: The sentences about using the term “RDT“ is confusing. The authors should clearly describe their definition of “RDT“. 

In response to the comments, we added the following sentences.

(Discussion, Page 8, Line 232-238)

Radiodynamic therapy (RDT) may be confused with photodynamic therapy (PDT) in appearance, but I use the term “RDT” intentionally. We are not the first to use this term; “RDT” was previously used to mean RT using the drug for PDT, but no direct data on the physicochemical reaction between ionizing radiation and sensitizers has been reported [26,27]. Therefore, we defined RDT as using a radiomediator, such as PpIX, which physicochemically enhances ROS generation by ionizing radiation based on experimental verification [15].

Reviewer 2 Report

The paper investigates the effect of 5-ALA administration on the fractionated radiotherapy in glioblastoma as well looks into safety of the experiments. The experiments were performed on mice models and cell line assays. The effects of treatment method was evaluated by measurement of PpIX concentration and tumor volume (using a caliper) up to 72 days. The results showed a decrease in the tumor volume when 5-ALA was used together with the fractionated x-ray radiation. The paper is well written, the study design is appropriate and, with the exception of a few unclarities, the methods are well described. The paper refers to some of the earlier publications of the authors, however, it does not sufficiently motivate the novelty of the study. Figures and tables are clear and easy to read. The discussion adequately addresses different aspects of the study and the future clinical aspects of the method. Overall, the paper is interesting and of a high scientific quality. There are some text similarities specifically in the material and methods with the authors’ earlier publications that should be revised to the possible extent.  

Detailed comments:  

Introduction:  

  • Page 2, paragraph 2, authors mention 5-ALA PDT application only for skin cancer while 5-ALA is also used for treatment of brain tumors which can be mentioned and cited here.   
  • Note that production of reactive oxygen occurs under laser light with a wavelength suitable for excitation of PpIX and not any laser light.  
  • Please motivate the novelty of the paper and include state of the art on this specific topic.  
  • The aim mentions only mice as testing platform but results (2.1.5) are also presented for the cellular studies using assays. Please explain consistently throughout the manuscript including the abstract.  

Material and methods: 

  • The paper can benefit from a short explanation at the start of this section regarding what platforms (e.g. cell culture/assays and mice) were considered for experiments.  

Results: 

  • It is more logical to first write about the material and methods, then results followed by discussion (the position of these sections does not follow this order in this manuscript)  
  • Page 5, line 7, correct ‘om’  
  • Section 2.4  The results obtained on the normal brain are expected since 5-ALA does not pass the blood brain barrier in the normal brain. Evaluation in the skin is more relevant, however in humans ALA is cleared from the skin within 1-2 days. How long after the ALA administration was the evaluation on the mice skin performed?  Please mention in the manuscript.
  • Page 7, line 2, please correct grammar  
  • Page 7, last sentence at the end paragraph is incomplete  

Discussion:   

  • If the PpIX produces reactive oxygen that positively affects the radiotherapy, it can be assumed that it absorbs the x-ray similar to the laser light that is used for PDD or PDT. Do the authors have any motivation for the underlying mechanism of the observed phenomenon that can be included in the discussion? 
  • Could the authors clarify how 240 mg/kg implemented in this study is considered equivalent to the human dose (20 mg/kg)? 

Author Response

First of all, we would like to express our cordial thanks for the constructive comments received from this reviewer.

The paper investigates the effect of 5-ALA administration on the fractionated radiotherapy in glioblastoma as well looks into safety of the experiments. The experiments were performed on mice models and cell line assays. The effects of treatment method was evaluated by measurement of PpIX concentration and tumor volume (using a caliper) up to 72 days. The results showed a decrease in the tumor volume when 5-ALA was used together with the fractionated x-ray radiation. The paper is well written, the study design is appropriate and, with the exception of a few unclarities, the methods are well described. The paper refers to some of the earlier publications of the authors, however, it does not sufficiently motivate the novelty of the study. Figures and tables are clear and easy to read. The discussion adequately addresses different aspects of the study and the future clinical aspects of the method. Overall, the paper is interesting and of a high scientific quality.

Thank you for your high evaluation for our paper.

There are some text similarities specifically in the material and methods with the authors’ earlier publications that should be revised to the possible extent. 

In response to the reviewer’s detailed comments, we revised our manuscript. We also confirmed by iThenticate which provide plagiarism detection service.

Detailed comments: 

Introduction: 

  • Page 2, paragraph 2, authors mention 5-ALA PDT application only for skin cancer while 5-ALA is also used for treatment of brain tumors which can be mentioned and cited here.

In response to the reviewer’s comments, we added the following sentences and references.

(Introduction, Page 2, Line 65-68)

For glioma, interstitial PDT (a stereotactic phototherapy using fiber optics as a laser diffuser due to limited light transmission) or intraoperative PDT (which uses a lighting device placed inside the surgical cavity) have been developed [8-10]. Clinical studies and clinical trials of 5-ALA PDT for non-dermatologic cancer are ongoing [11,12].

  1. Stummer, W.; Beck, T.; Beyer, W.; Mehrkens, J. H.; Obermeier, A.; Etminan, N.; Stepp, H.; Tonn, J. C.; Baumgartner, R.; Herms, J.; Kreth, F. W. Long-sustaining response in a patient with non-resectable, distant recurrence of glioblastoma multiforme treated by interstitial photodynamic therapy using 5-ALA: case report. J Neurooncol. 2008, 87(1), 103–109.
  2. Johansson, A.; Faber, F.; Kniebühler, G.; Stepp, H.; Sroka, R.; Egensperger, R.; Beyer, W.; Kreth, F. W. Protoporphyrin IX fluorescence and photobleaching during interstitial photodynamic therapy of malignant gliomas for early treatment prognosis. Lasers Surg Med. 2013, 45(4), 225–234.
  3. Dupont, C.; Vermandel, M.; Leroy, H. A.; Quidet, M.; Lecomte, F.; Delhem, N.; Mordon, S.; Reyns, N. INtraoperative photoDYnamic Therapy for GliOblastomas (INDYGO): Study Protocol for a Phase I Clinical Trial. Neurosurgery. 2019, 84(6), E414–E419.
  4. Casas A. Clinical uses of 5-aminolaevulinic acid in photodynamic treatment and photodetection of cancer: A review. Cancer Lett. 2020, 490, 165–173.
  5. Lietke, S.; Schmutzer, M.; Schwartz, C.; Weller, J.; Siller, S.; Aumiller, M.; Heckl, C.; Forbrig, R.; Niyazi, M.; Egensperger, R.; Stepp, H.; Sroka, R.; Tonn, J. C.; Rühm, A.; Thon, N. Interstitial Photodynamic Therapy Using 5-ALA for Malignant Glioma Recurrences. Cancers (Basel). 2021, 13(8), 1767.

  • Note that production of reactive oxygen occurs under laser light with a wavelength suitable for excitation of PpIX and not any laser light.

In response to the reviewer’s comments, we added the words.

(Introduction, Page 2, Line 70)

The mechanism of 5-ALA-PDT is as follows: exogenous 5-ALA induces the accumulation of protoporphyrin IX (PpIX) in cancer cells, PpIX produces reactive oxygen species (ROS), mainly 1O2, under laser light suitable for excitation of PpIX, which induces cellular damage and death [13,14].

  • Please motivate the novelty of the paper and include state of the art on this specific topic.

In response to the reviewer’s comments, we added the following sentences and references.

(Introduction, Page 2, Line 80-84)

Due to advances in glioblastoma prevention, early detection, and treatment, 5-year glioblastoma survival has increased from 4% to 7% over the past four decades [18]. The development of treatment methods that aim to prolong life and cure the condition is strongly desired. In a retrospective analysis, RT improved the overall survival of glioblastoma patients with poor performance status [19]. We believe that improving RT is a key issue in controlling glioblastoma.

  1. Miller, K. D.; Ostrom, Q. T.; Kruchko, C.; Patil, N.; Tihan, T.; Cioffi, G.; Fuchs, H. E.; Waite, K. A.; Jemal, A.; Siegel, R. L.; Barnholtz-Sloan, J. S. Brain and other central nervous system tumor statistics, 2021. CA Cancer J Clin. 2021, 10.3322/caac.21693.
  2. Schröder, C.; Gramatzki, D.; Vu, E.; Guckenberger, M.; Andratschke, N.; Weller, M.; Hertler, C. Radiotherapy for glioblastoma patients with poor performance status. J Cancer Res Clin Oncol. 2021, 10.1007/s00432-021-03770-9.

  • The aim mentions only mice as testing platform but results (2.1.5) are also presented for the cellular studies using assays. Please explain consistently throughout the manuscript including the abstract.

Thank you for the comment. In response to the reviewer’s comments, we added the following sentences in introduction section. We didn't change the abstract, because the main subject of our paper is to study RDT for glioblastoma fractionated radiotherapy, and there is the tight character limit of the abstract section.

(Introduction, Page 2, Line 84-88)

In this study, we examined the cellular response to single-dose irradiation with 5-ALA for two types of glioblastoma and the effect of RDT for the standard fractionated RT treatment protocol for glioblastoma after surgery using the same two types of glioblastoma-bearing mice. Then, we verified the safety of 5-ALA combined with standard fractionated RT in normal mic

Material and methods:

  • The paper can benefit from a short explanation at the start of this section regarding what platforms (e.g. cell culture/assays and mice) were considered for experiments.

Thank you for the comment. In response to the reviewer’s comment, I shortened the subtitle, added short explanations as follows and divided to the paragraphs.

(Materials and Methods, Page 11, Line 339-362, Page 12, Line 388-407)

4.4. Subcutaneous implantation of U251MG and U87MG glioma cells and X-ray irradiation in mice

->

4.4. RDT treatment for two types of glioblastoma-bearing mice

Subcutaneous implantation of U251MG and U87MG glioma

Grouping and RDT fractionated irradiation

Evaluation during the irradiation period

4.7 Safety test of 5-ALA on fractionated irradiation

->

4.7. Safety test of 5-ALA on fractionated irradiation

Mice and breeding condition

Grouping and RDT fractionated irradiation

Evaluation after RDT fractionated irradiation

Results:

  • It is more logical to first write about the material and methods, then results followed by discussion (the position of these sections does not follow this order in this manuscript)

Thank you for the comment. However, there is the template for the IJMS, and the order is results, discussion, and materials and methods, so I followed the template order.

  • Page 5, line 7, correct ‘om’

We corrected it.

(Results, Page 5, Line 161 in the revised version)

  • Section 2.4 The results obtained on the normal brain are expected since 5-ALA does not pass the blood brain barrier in the normal brain. Evaluation in the skin is more relevant, however in humans ALA is cleared from the skin within 1-2 days. How long after the ALA administration was the evaluation on the mice skin performed?  Please mention in the manuscript.

Thank you for the comment. In response to the reviewer’s comment, we added the safety test protocol as figure 4 and add following sentences and references.

(Results, Page 7, Line 188-199)

After the completion of 60 Gy fractional irradiation combined with 5-ALA treatment, pathological evaluation of the irradiation site of the skin and brain was performed. Blood was collected and tissues of the skin and brain were fixed 24 h after the final 5-ALA administration. The safety test protocol is illustrated in Figure 4.

Pathological evaluation of the skin of all X-ray irradiated mice revealed the following: (1) hypertrophy (mild to moderate) of epidermal or epidermal cells, (2) hypertrophy (mild) of hair follicle epithelial cells, (3) apoptosis (mild) of epidermal cells, (4) atrophy/disappearance (mild) of hair follicles, and (5) atrophy/disappearance (moderate) of epidermal cells. However, these skin lesions were not different among the X-ray irradiation groups due to 5-ALA administration (Figure 5 and Table 3). Pathological evaluation of the brain showed no changes associated with X-ray irradiation or 5-ALA administration (Table 3). In the normal brain, 5-ALA does not pass through the blood brain barrier [20, 21], which is consistent with our pathological examination results.

  1. García, S. C.; Moretti, M. B.; Garay, M. V.; Batlle, A. Delta-aminolevulinic acid transport through blood-brain barrier. Gen Pharmacol. 1998, 31(4), 579–582.
  2. Terr, L.; Weiner, L. P. An autoradiographic study of delta-aminolevulinic acid uptake by mouse brain. Exp Neurol. 1983, 79(2), 564–568.

  • Page 7, line 2, please correct grammar

We corrected it.

(Results, Page 7, Line 192 in the revised version)

Pathological evaluation of the in skin of all X-ray irradiated mice revealed the following:

  • Page 7, last sentence at the end paragraph is incomplete

We corrected the sentence as follows.

(Results, Page 7, Line 197-198 in the revised version)

Pathological evaluation of the brain showed no changes associated with X-ray irradiation or 5-ALA administration (Table 3).

Discussion:  

  • If the PpIX produces reactive oxygen that positively affects the radiotherapy, it can be assumed that it absorbs the x-ray similar to the laser light that is used for PDD or PDT. Do the authors have any motivation for the underlying mechanism of the observed phenomenon that can be included in the discussion?

Thank you for the comment. In response to the reviewer’s comment, we have also added the following text.

(Discussion, Page 8, Line 226-232)

Previously, we attempted to detect the physicochemical reaction between PpIX and X-rays using ROS indicators [15]. To identify the ROS, 2-[6- (4-amino)phenoxy-3H-xanthen-3-on-9-yl] benzoic acid (APF), which mainly detects ∙OH, and dihydroethidium (DHE), which detect O2∙− and/or 1O2, were used together with ethanol as an ∙OH quencher. The results indicated that PpIX contributes to enhanced generation of ∙OH, O2∙−, and/or 1O2 under X-ray irradiation, but to O2∙− and/or 1O2 under UV irradiation. PpIX was excited by UV light. Although the types of ROS produced are different, it has been shown that a reaction similar to photoexcitation is triggered by X-rays [15].

  • Could the authors clarify how 240 mg/kg implemented in this study is considered equivalent to the human dose (20 mg/kg)? Response

Thank you for the comment. In response to the reviewer’s comment, we have also added the following sentences and reference

(Discussion, Page 10, Line 303-310)

We utilized a guidance document from the US Department of Health and Human Services to determine dosages, which was the Food and Drug Administration's (FDA) "Industry Guidance for Estimating the Maximum Safe Starting Dose in Initial Clinical Trials for Healthy Adult Volunteer Therapeutics" [49]. FDA guidance recommends to evaluate the available animal data so that a "Non Observed Adverse Effect Level (NOAEL)" can be determined and calculate human equivalent dose (HED). For mice, HED multiplies the animal dose by 0.08 [49]. Therefore, 240 mg/kg is 19.2 mg/kg for humans, which is roughly equivalent to the human dose of 20 mg/kg for PDD.

  1. US Food and Drug Administration. Estimating the Maximum Safe Starting Dose in Initial Clinical Trials for Therapeutics in Adult Healthy Volunteers. US Food and Drug Administration; 2005, 1-27.  

Round 2

Reviewer 2 Report

The authors have replied to all my of my comments and implemented appropriate changes in the manuscript.